# How Do Location-Based Augmented Reality Games Improve Physical and Mental Health? Evaluating the Meanings and Values of Pokémon Go Users’ Experiences through the Means-End Chain Theory

**DOI:** 10.3390/healthcare9070794

**Published:** 2021-06-24

**Authors:** Gordon Chih-Ming Ku, I-Wei Shang, Meng-Fan Li

**Affiliations:** 1Department of Sport Management, National Taiwan University of Sport, Taichung City 404, Taiwan; GordonKu@gm.ntus.edu.tw; 2Department of Physical Education and Kinesiology, National Dong Hwa University, Hualien County 974, Taiwan; alanlily0727@yahoo.com.tw

**Keywords:** soft laddering method, social relationships, augmented reality, online games, Pokémon Go

## Abstract

New technology has dramatically changed online games and blurred the boundary between active and passive activities. This study aims to explore the meanings and values of augmented reality online games by examining users’ Pokémon Go experiences through the means-end chain theory. Using data from interviews with 34 Pokémon Go users, this study adopts the soft laddering method to identify Pokémon Go’s potential attributes, consequences, and values, and to construct a hierarchical value map. The results indicated that Pokémon Go users pursue social relationships through play, and these relationships are triggered by the benefits of making new friends, maintaining current relationships with friends and family, and the attributes of prevalence, childhood memory, game design, and augmented reality. Subsequently, this study describes how Pokémon Go can be considered an active leisure activity because of its social, mental, and physical benefits and assesses the implications of its findings.

## 1. Introduction

Numerous studies have verified that online games can negatively influence peoples’ well-being [1,2]. Männikkö, Mendes, Barbosa, and Reis [1] systematically reviewed the relationship between health determinants and playing video games, and their results indicated that playing video games online is highly related to risk behaviors and adverse overall health outcomes. However, location-based augmented reality (AR) technology has brought gamers a whole new interface, playing games, and a set of benefits. AR subverts the perception that gaming is a passive leisure activity because it enables users to play via their smartphones, play outdoors, pursue social connections, and navigate and perceive the real world while they play [3,4]. Jang, Ko, Lee, and Kim [5] mentioned that AR technology, when applied to e-leisure activities, could increase users’ quality of life, sense of self-worth, and enjoyment, and AR online games generate more positive and active outcomes.

Pokémon Go is one of the most popular location-based AR games in the world. As part of the hugely popular Pokémon series of games, it integrates users’ locations via Global Positioning System (GPS) technology, view (via their camera), and virtual information—the various wandering creatures of the Pokémon universe—with users’ real, immediate environment [6]. Pokémon Go made USD 207 million in its first month and was downloaded over 100 million times by users across 30 countries in its first few weeks [7,8]. Currently, it boasts over one billion downloads and five million active players per day and generates USD 3 billion per year [9]. Part of Pokémon Go’s incredible success stems from how it combines physical, mental, emotional, and social experiences [6]. Therefore, this study selected Pokémon Go as its subject to explore the value chain of arguably the most popular, successful, and potential-laden location-based AR game.

This study assesses users’ experiences and perceptions of Pokémon Go’s values to understand the potential contributions to leisure in general. It applies means-end chain (MEC) theory to more deeply understanding users’ leisure experiences. This theory provides a cognitive structure that illustrates individuals’ leisure experiences on a spectrum encompassing the attributes of that activity and users’ values. This study’s findings help explain how individuals choose leisure activities, develop various strategies for applying potential educational and therapeutic benefits, and identify whether Pokémon Go is an active or passive leisure activity. In short, this study has four specific objectives: first, to identify the attributes of Pokémon Go; second, to find the consequences of each attribute; third, to identify the values of Pokémon Go, which are triggered by consequences or attributes; and finally, establish hierarchical connections among attributes, consequences, and values and construct a hierarchical value map (HVM) for Pokémon Go users’ experiences.

## 2. Literature Review

### 2.1. Online Games and Leisure

The Internet is a contemporary technology that creates a new, deeply interconnected virtual space for human interaction and activity. It has fundamentally changed peoples’ concepts and experiences of leisure [10]. Several studies have been devoted to exploring peoples’ motivations and the various consequences of playing online games. Common motivations include performance, enjoyment, achievement, social interaction, flow experience, advancement, mechanics, entertainment, personal identity, and escapism [11,12,13,14]. In addition, users’ motivations can be categorized into functional (i.e., social interaction, entertainment, etc.) and emotional (i.e., enjoyment and achievement) motivations [14].

Online games allow users to interact via multiplayer games in which users can play with other users worldwide [10,13,15]. This kind of social interaction distinguishes online games from traditional video games [16]. Trepte, Reinecke, and Juechems [17] found that online gamers’ social proximity and mutual familiarity can affect their online bridging and bonding social capital and extend to offline activities. Online gamers can receive social support, make new friends, and communicate and interact with other players in virtual space, and these interactions can contribute to gamers’ psychological health and social needs. However, as mentioned above, doubts regarding whether online games positively affect users’ well-being—especially their physical well-being—persist. 

Online gamers are sedentary for long periods, especially if they often play at a high level of play. Griffiths, Davies, and Chappell [18] assessed 11,457 online gamers to establish benchmark data for further study. They found that 24% of these gamers played online games for more than 41 h per week—approximately 6 h per day. Numerous studies have indicated that lengthy immersion in online games can negatively affect individuals’ psychological and social health. Studies have shown that such immersion can reduce gamers’ self-efficacy and self-esteem and make them more likely to suffer anxiety, lose track of time, drop out of school, quit their jobs, and distance themselves from their friends and family [19,20,21,22]. Further, certain studies have indicated that such immersion can have adverse effects on gamers’ physical health as well [15,23]. Kim and Kim [15] developed a multi-dimensional scale of problematic online game use and perceived that deeply immersed gamers had worse overall health, frequent headaches, and increasingly serious eyesight problems. In short, there are several ongoing arguments about whether online gaming’s overall effects and consequences should be described as positive or negative. 

Furthermore, several studies have adopted theories to explain gamers’ online behavior. These theories include the theory of planned behavior [24], use and gratification theory [13], technology acceptance model [25], flow theory, and humanistic needs theory [26]. Each of these theories focuses on the causal relationship between antecedents, intentions, and behaviors. For example, Merhi [13] applied use and gratification theory to explore gamers’ usage intention while playing online games and found that enjoyment, achievement, and social interaction were the three significant antecedents of their usage intentions. Alzahrani, Mahmud, Ramayah, Alfarraj and Alalwan [24] used the theory of planned behavior and several potential variables to construct and verify a model of gamers’ online behavior. They and found that gamers’ actual behavior was influenced by their flow experience, perceived enjoyment, attitude, subjective norms, and perceived controls on their behavior. These theories have offered reliable ways to understand why people play games online, but there is still a lack of studies on the attributes, consequences, and personal values of online gaming, especially with regarding to new and emerging technology such as location-based AR. 

### 2.2. Pokémon Go and Leisure Activity

AR technology enhances users’ surrounding environment with virtual information, which is rendered in 3D through users’ smartphones, and layered atop the real world to create the appearance of co-existence [27]. AR technology has been developing rapidly; in recent years, traditional AR devices, such as a head-mounted display, have been replaced by smartphones that can integrate all the essential AR technologies and functions [28]. For example, smartphones’ geolocation capabilities can immediately and faithfully render context-aware, interactive virtual environments, even as users move around [28,29]. As AR technology becomes more accessible, location-based AR games have emerged as a new type of leisure activity. Fischer [30] defines location-based AR games as a link of media- and outdoor-time within the devotion of leisure-time (p. 23). For this reason, the main differentiation between traditional online games and location-based AR games is the fact that the latter allows players to participate in gaming experiences outdoors.

Previous studies have examined Pokémon Go users’ motivations and outcomes [6,8,31,32,33]. Several studies have extended existing frameworks to construct a Pokémon Go game-specific motivation scale. For example, Zsila, Orosz, Bőthe, Tóth-Király, Király, Griffiths, and Demetrovics [8] combined qualitative and quantitative methods to extend the Motives for Online Gaming Questionnaire and develop a motivation scale for Pokémon Go. The study identified ten dimensions of motivation—social, escapism, competition, coping, skill development, fantasy, recreation, outdoor activity, nostalgia, and boredom. In addition, several studies have focused on Pokémon Go users’ motivation. Caci, Scrima, Tabacchi, and Cardaci [6] developed their Pokémon Go Motivational Scale through a systematic literature review. The scale consisted of 13 items: curiosity, creativity, different cultures, new people, friends, social experiences, free time, trendiness, escapism, aggression, sexuality, personality, and physical activity. These motivation scales show that Pokémon Go shares some attractive features of traditional online games but is unique in motivating users to get outdoors and engage in physical activity.

Pokémon Go has been proven to bring users mental, social, and physical benefits [6,31,34,35,36,37]. Tong, Gupta, Lo, Choo, Gromala, and Shaw [31] argued that Pokémon Go could encourage people to go outdoors, engage in positive physical and emotional activities, and explore their surroundings. Many empirical studies have provided evidence that Pokémon Go can effectively promote users’ physical activity and outdoor activity [6,34,38]. For example, Nigg, Mateo, and An [34] found that Pokémon Go significantly increased users’ vigorous physical activity by 50 min per week and reduced their sedentary activities by 30 min per day. However, there has been a distinct lack of studies that have focused on the specific connections between attributes, consequences, and values in Pokémon Go. 

As Baranowski [39] proposed, Pokémon Go has potentially powerful ways to increase physical activity. In order to design a more appealing and longer-lasting program, the characteristics and the users’ experiences should be further understood. By exploring these relationships, we can not only clarify why people play Pokémon Go, what benefits they receive from playing, and what players value about these games—we can also deepen our understanding of Pokémon Go apply this understanding to unlock the educational and therapeutic potential of Pokémon Go. 

### 2.3. Means-End Chain Theory in Leisure Research

MEC theory can explain how a leisure activity helps someone achieve a desired end state. The cognitive structure employed by MEC theory consists of three levels of abstraction—attributes, consequences, and values—each of which is further divided into two different categories [40]. Attributes refer to the features of leisure products or services perceived by users [41]. They are divided into tangible (i.e., game and activity) and intangible (i.e., experiences and memories) characteristics of leisure activities. Consequences refer to users’ experiences of leisure products or services [42], which include functional (or physical) and psychological consequences [40]. Value refers to users’ innermost personal desires for joining a leisure activity. In the MEC schema, values are further divided into instrumental and terminal values, which represent individuals’ preferable modes of conduct (i.e., achievement and capability) and ultimate ideal state (i.e., happiness or fulfillment), respectively [40,41]. Value has been invoked as a valuable predictor of personal behaviors and attitudes [43,44].

MEC theory has previously been used to model and explore individuals’ consumption behavior—how individuals choose products or services to satisfy their needs and desires. For instance, it has been utilized to develop marketing strategies by other studies [45]. In addition, MEC theory offers a structure of means to desired ends to help researchers understand the relationship between personal decision-making and cognitive structures [40]. Means refers to personal perceptions of a product or service’s attributes; end represents the values that an individual desires [41]. MEC theory has been used to investigate a wide variety of sub-topics within leisure and tourism studies, such as travel and leisure motivation [42,43,46], destination choice [47], video-sharing websites [40], virtual reality [41], the psychological values of recreational cyclists [43] and travel and leisure experience [40,48,49]. For example, Lin, Jeng, and Yen [41] applied MEC theory to explore elderly peoples’ awareness, decision-making procedures, and personal values when selecting virtual reality (VR) leisure activities. The results indicated that good memories are the terminal value that leads to a different awareness of VR leisure activities and different decision-making processes. Therefore, this study’s use of MEC theory will provide insights into Pokémon Go users’ inner cognitive structure towards this specific leisure activity and a fully comprehensive picture of their leisure experiences with the game. 

## 3. Methodology

### 3.1. Research Design

The laddering technique is an essential approach in MEC theory. This technique is used to extract abstractions such as attributes, consequences, and values via in-depth interviews [42]. The technique can be divided into soft and hard laddering approaches. The former adopts a qualitative method—open-ended questions and in-depth one-on-one interviews—in order to collect unrestricted, rich, qualitative, personally meaningful information [41,42]. The latter employs structured questionnaires to collect data on attributes, consequences, and values via telephone, email, the Internet, or self-administered surveys [38]. To better explore Pokémon Go users’ perspectives, this study adopted the soft laddering technique to collect deeply qualitative reflections from users. 

The adult users who have been playing Pokémon Go for at least one year were invited to participate in this study. All participants volunteered to accept a one-on-one interview invitation. Before starting the interview, informed consent was provided to explain to participants about the interview procedure’s risks, benefits, and alternatives. The participants’ signatures were requested if they completely understood their rights in the interview process and voluntarily participated in the study. Afterward, the interviewees followed the interview guide to probe the Pokémon Go users’ experiences. A recorder recorded the interview content. 

The interview questions used in this study were developed by considering previous studies. Interviews began with three open-ended questions. First, “Why do you like to play Pokémon Go?”, led interviewees to express their perceptions of the game’s essential attributes. Second, “What consequences can this attribute bring to you?”, probed interviewees’ perceptions of the potential benefits and costs of each of the game’s attributes. Third, “What personal values were achieved through the consequence?”, saw interviewees identify the values they achieved from each consequence. In addition, the interviewers continuously asked the interviewees, “Why is it important to you?” after interviewees expressed a fundamental abstraction and systematically probed rich information from interviewees until they were no longer able to answer. Interviews were recorded, and afterward, interviewees were required to report their gender, age, occupation, and estimate how much time they spent using Pokémon Go per day.

Soft laddering is conducted in three steps: identifying attributes, building linkages between abstractions, and drawing the hierarchical value map (HVM) [41,49,50]. In this study, first, participants were asked to express what they felt were the critical attributes of Pokémon Go. Subsequently, they were asked to explain how they perceived the relationship between consequences and personal values vis-à-vis their experience of Pokémon Go. The frequency of the linkages between abstractions was calculated into an implication matrix. Finally, we created the HVM to illustrate the various hierarchical linkages between attributes, consequences, and values in Pokémon Go users’ experiences. 

### 3.2. Eligible Participants

In order to enhance the validity of data in this study, the authors employed purposive and snowball sampling to select 34 eligible study participants. Eligible participants have been continuously playing Pokémon Go for over one year, be at least 20 years old, and voluntarily submit themselves to the 30–40 min interviews described above. According to the above principles, eligible participant recruitment was divided into two stages. In the first stage, 10 and 13 eligible participants were recruited from National Don Hwa University and Qixingtan Beach, respectively—two popular spots for Pokémon Go activity in Hualien County, Taiwan. In the second stage, the remaining 11 eligible participants were recruited from the initial participants’ acquaintances. Interviews were conducted face-to-face or online depending on the interviewees’ preferences. The sample size of the study was decided by data saturation. The data collection was stopped at number 34 participants because no additional data could be established to develop new properties of categories [51].

Table 1 displays the demographic information for this study’s 34 participants; of these, 61.8% were male and 38.2% were female. Most participants were 20–25 years old (32.3%) or 26–30 years old (29.4%). Around one-third were laborers (35.5%), and another third were students (32.3%). Most participants played Pokémon Go for either less than 60 min (38.2%) or 121–180 min (38.2%) each day.

### 3.3. Ethical Considerations

Ethical consideration was conducted in four steps to ensure that the interviewees’ information was protected. First, the researcher provided informed consent to each interviewee. Second, interviewers exhaustively interpreted the study’s purpose and the interview procedure to participants before interviews. Third, the rights of interviewees were clearly declaimed—including interviewees’ right to refuse answers, end the interview, quit the study, and have researchers obscure any identifying information they might provide. Fourth, interviewees were asked to sign informed consent forms to signal that they agreed to the study’s principles and had participated in the study voluntarily. 

### 3.4. Data Analysis

The collected data was analyzed in three steps, according to MEC theory. In the first step, the researchers performed a content analysis to elicit terms (codes) from interview transcripts. They subsequently classified similar terms into different levels of abstraction. This was done by three coders (one professor, one lecturer, and one graduate student) who had passed qualitative method training and were familiar with the location-based AR game issue and the MEC theory. Subsequently, the key terms were named and classified based on the framework in Lin and Fu [40]; however, coders were permitted to find or create new terms from the interview transcripts. Inter-coder reliability was 0.92, indicating that the content analysis was highly consistent and reliable. 

In the second step, an implication matrix was developed to uncover hierarchical linkages between attributes, consequences, and values. Reynolds and Gutman [50] mentioned that the ability to uncover the relationships between terms is the most crucial application of MEC theory. Therefore, in this study, terms with high frequency were eliminated if they did not display any relationships to other terms. The implication matrix revealed the linkages between terms and calculated the number of times each term connected to each other term. This gave the researchers the data necessary to display significant linkages via the HVM. 

In the third and final step, the researchers constructed the HVM—a tree diagram that shows Pokémon Go users’ inner thought processes by displaying the various linkages between different levels of abstraction. A cut-off point of 5% of the sample size was chosen, which has been widely accepted in previous studies as a standard way to identify the available connections between abstractions [49]. However, as per Lin et al. [41], a frequency of linkage between particular abstractions of three to five was also accepted as a cut-off point for identifying the available connections. Researchers evaluate their databases first and then choose an appropriate cut-off point in order to provide ‘the most informative and most stable set of relations [50]. In order to effectively simplify and visualize the HVM, this study considered all connections with a frequency of linkage above three as available.

## 4. Results

The results of this study have three primary items and themes: presentation of the abstractions of Pokémon Go users’ experiences, presentation of the overall HVM, and three sub-patterns of Pokémon Go users’ experiences, and interpretations of the meaningful values derived from users’ Pokémon Go experiences and the influences these values have on users’ decision-making processes.

### 4.1. The Abstractions of Pokémon Go Users’ Experiences

Twenty-one abstractions were extracted via a content analysis of the 34 participant interviews (Table 2). The abstractions included six attributes, ten consequences, and five values. The concrete attributes included prevalence (31), game design (24), and augmented reality (19), while the abstract attributes included childhood memories (28), novelty (15), and curiosity (7). These findings are similar to that of Tong et al. [31], Pokémon Go’s location-based AR technique, game design, and users’ nostalgia for the Pokémon series of games and shows motivated individuals to engage in the game. Thus, trends in technology and childhood memories or nostalgia can be regarded as the main tangible and intangible attributes of users’ experiences, respectively. 

The ten consequences were categorized into three psychological consequences—better mood (17), fantasy (9), and nostalgia (8)—and seven physical consequences—maintaining relationships with friends and family (30), making new friends (29), skill development (16), going outdoors (16), physical activity (15), re-acquainting environment (10), and competition (10). The findings are similar to empirical studies that have proven that Pokémon Go can provide users with various benefits [6,8,31,34]. For instance, an empirical study conducted by Caci et al. [6] showed that Pokémon Go could bring social benefits, satisfy personal needs, and increasing physical activity in players. A recent study showed social benefits to be the most significant advantage of playing Pokémon Go.

Further, this study identified three instrumental values—social relationship (26), people–place relationship (16), and personal health (15)—and two terminal values—a sense of accomplishment (17) and self-fulfillment (13). Social relationships are the essential value of AR, which builds or rebuilds the connections between families, relatives, old friends, new friends, and the players themselves. Tom Dieck and Jung [52] mentioned that AR games could enhance social interaction between users and provide them with some measure of social fulfillment. Thus, location-based AR technology can help redefine video games and online games as inherently social activities, which have the potential to produce rather than restrict positive social behavior and outcomes. 

### 4.2. Hierarchical Value Map for Pokémon Go Experiences

The implication matrix (Table 3) for Pokémon Go users’ experiences was used to identify the various relationships between attributes, consequences, and values in this study. The matrix displays the frequency of direct and indirect linkages between attributes, consequences, and values. This study recorded a total of 299 direct linkages and 60 indirect linkages. In Figure 1, a line’s thickness represents the number of linkages, which shows the different levels of significant connections. For example, the thickest lines represent more than ten linkages, the next-thickest line between six and nine linkages, and the thinnest line less than five linkages. Using this matrix, the authors then constructed the HVM for Pokémon Go users’ experiences, displaying 25 total linkages—four powerful connections, seven strong-to-average connections, and 14 fewer strong connections. The overall HVM is separated into three sub-themes to illustrate Pokémon Go users’ experiences.

### 4.3. Prevalence and Childhood Memory Contribute to Social Relationships

Pokémon Go’s prevalence and popularity trigger individuals to download and play the game. Prevalence (A1) is linked to social benefits, such as maintaining connections with friends and family (C1) and making new friends (C2). These two benefits are rewarded with social relationships (V1). Pokémon Go’s incredible popularity means many users talk about their experiences playing the game with their friends and family. One participant stated, “*I play Pokémon Go because all my family and friends are engaging in this game.*
*Usually, my family watches television or does their thing*
*after dinner. Now, we*
*take a walk and [play*
*the game together]*
*after meals**. I feel that [our]*
*family relationship is closer than before*”. Some participants reported that they made new friends through the game, and Pokémon Go increased their social ability and desire to interact with strangers. One participant mentioned that “*while I was previously timid and not very social, I felt that the game improved my social skills, and now I do not feel afraid to interact with people and make a few new friends*”. 

As users play Pokémon Go for nostalgic reasons, childhood memories (A2) can help them maintain connections with friends and family (C1), improve their mood (C3), enjoy competition (C8), and feel nostalgia (C10). Maintaining relationships with family and friends leads players to attain the value of having social relationships (V1). Pokémon Go has successfully reminded players of their delightful childhood memories of the cartoon, which contributes to them sharing these experiences with their siblings and friends. In this vein, one participant reflected, “*I invited my little brother to play Pokémon Go together because it was our favorite cartoon in childhood. Now, we talk more frequently because of Pokémon Go*”.

### 4.4. Benefits of Pokémon Go’s Game Design and Augmented Reality

As mentioned above, Pokémon Go’s use of AR technology and design as an outdoor activity distinguishes it from traditional online games. This study found that the game’s design (A3) and incorporation of AR technology (A4) led to various consequences, including maintenance of connections with friends and family (C1), making new friends (C2), developing skills (C4), increasing users’ outdoor activity (C5) and participation in physical activity (C6), re-learning about their surrounding environment (C7), and enjoying competition (C8). The AR technology and game design bring people together by encouraging friends and family relationships, new friendships, outdoor activity, and re-acquainting one’s environment. Thus, Pokémon Go seems to dramatically reshape general impressions of online games as passive, isolated, and largely sedentary activities. 

These benefits are accompanied by each of the values identified in this study: users’ social relationships (V1), users’ sense of accomplishment (V2), reflections on the relationship between people and places (V3), improvements in users’ personal health (V4), and self-fulfillment (V5). When Pokémon Go users play together, they can upgrade their in-game skills and capture more powerful and rare monsters, fostering a sense of achievement. One user reflected, *“**Being a Pokémon trainer was*
*my dream when I was a child. Pokémon Go makes this*
*dream come true. Therefore, I would gain a great feeling of*
*achievement if I could*
*become a powerful*
*trainer and capture all the*
*monsters in the game”.* Also, Pokémon Go helped users go outdoors more frequently and learn more about their immediate environment, which led several players to feel better physically and acquaint their surrounding environment. As a participant stated, *“I start to take a walk around my home due to play Pokémon Go. Suddenly, I realize that I am very unfamiliar with the environments near my home. For example, some of the [in-game] temples took me to places I had never been to in my daily life. Now I walk over five kilometers every day while playing, which has helped me engage with my neighborhood and improve my physical stamina*”.

### 4.5. Escaping from Daily Routines: Curiosity and Novelty

The novelty of AR technology and peoples’ curiosity about it satisfies several Pokémon Go users’ desire for a fresh escape from reality and daily routines. Attributes such as the game’s novelty (A5) and users’ curiosity (A6) make them likely to fantasize about alternate realities while playing (C9) and increase their level of outdoor activity (C5). Location-based AR games allow players to adopt a new virtual identity in a familiar physical space, bringing some players improved health (V4) and self-fulfillment (V5). This is a quantum leap forward in online games’ effects on human health; although several studies have agreed that traditional online games can positively affect mental health [24,53], no such agreement exists regarding physical health. Therefore, the location-based AR technology of Pokémon Go increases the games’ potential for improving users’ physical health [6,34]. For instance, one participant stated that “*I have set a goal for how many monsters to capture per day. I will not stop if I do not achieve the goal. Sometimes I walk more than ten kilometers a day. Consequently, I have reduced my weight by a couple of kilograms in the last month*”. 

### 4.6. Meaningful Value Chains

The HVM for Pokémon Go users’ experiences identifies 11 value chains that can help us understand why users play Pokémon Go and which benefits and values are derived from which attributes (Figure 2). Most users play the game in pursuit of social relationships. Other studies have indicated that Pokémon Go’s game design boosts the likelihood of social interactions between users in their area [54]. This study found that nostalgia is vital to the game’s success—a finding which is borne out by other studies [8,35,55]. In short, this study found that the Pokémon brand’s success and the popularity of Pokémon Go have created a “social world” of their own and that interactions between people in this social world are facilitated by the game’s design and incorporation of location-based AR technology since Pokémon Go spread out on the world. Users share their experiences, knowledge, information, and skills with other users within the “Pokémon Go social world”. 

## 5. Conclusions and Study Implications

This study found that users value the social relationships that Pokémon Go facilitates between friends, family, strangers, and neighbors. This result is supported by a number of studies [6,35,55]. In an experimental study of the effects of Pokémon Go on adolescents’ cognitive performance and emotional intelligence by Ruiz-Ariza, Casuso, Suarez-Manzano, and Martínez-López [56]. This experimental group that accepted eight weeks of treatment with Pokémon Go significantly increased their sociability and improved their social relationships compared to the control group. Furthermore, Pokémon Go extends online social interactions offline, enriching both online and offline relationships. As a result of the game’s location-based AR technology making users visible to one another when they are in the same area, especially a public place, it can facilitate the creation of new bonds of friendship, community, and solidarity [35]. Therefore, Pokémon Go may be considered a novel form of therapy for socially withdrawn people or experience some barrier to social interaction. Tateno, Skokauskas, Kato, Teo, and Guerrero [57] used Pokémon Go as a psychotherapeutic tool for social withdrawal patients, and their results were largely positive. These kinds of applications are the future of Pokémon Go and other location-based AR games. 

In short, Pokémon Go brings various benefits to its users. Although, as mentioned, the literature has indicated that traditional online games can improve users’ mental health, Pokémon Go’s game design and incorporation of location-based AR technology mean that it provides more physical and social benefits to users than its traditional online counterparts. The location-based AR technology and game design of Pokémon Go have been shown to avoid the usual lack of physical activities and sedentary lifestyle associated with online games, as well as to increase outdoor physical activity [6,8,34,35]. Although Pokémon Go cannot replace physical exercise in general and receive sports benefits directly, it assists people in starting their physical activity and social interaction. Accordingly, this study proposes that Pokémon Go is not treated as pure entertainment, but as a potential tool for promoting public health, well-being, and outdoor exploration [54,55]. In short, it can be considered as an instrument of public well-being promotion [54].

Pokémon Go should be considered a social medium for environmental education because of its GPS function and virtual information. The present study confirms that Pokémon Go motivates users to acquaint or re-acquaint themselves with the nearby environment and landmarks, creating or rebuilding relationships between people and place. Several studies have demonstrated that Pokémon Go can encourage players to acquaint or re-acquaint nearby landmarks, explore their environments, move beyond their neighborhoods, develop a sense of community, and promote active learning in local neighborhoods [31,35,54]. Therefore, environmental education designers may consider utilizing Pokémon Go or other location-based AR games as a partial measure to facilitate motivations and enhance learning efficiency in environmental education programs. 

Finally, Pokémon Go blurs the already ambiguous boundary between online games as passive and active leisure activities. This study indicates that this boundary needs to be redefined to respond to new and emerging technologies and that Pokémon Go should perhaps lead the way in such redefinition as it has created its genre of active, location-based AR games leisure. 

## 6. Limitations of the Study

This study examined Pokémon Go users’ experiences and attempted to identify significant value chains among these experiences. However, it has a few limitations: first, this study is limited to Taiwanese users. Although other similar studies have been conducted in different contexts, the limitations of this study mean that it is difficult to generalize its results. Future studies should collect data from many more countries, including Western countries. Second, this study adopted the soft laddering method to construct HVM to model Pokémon Go users’ experiences. However, there is still a lack of quantitative data verifying the HVM’s structure. Therefore, the hard laddering method could be helpful in future empirical studies.

## Figures and Tables

**Figure 1 healthcare-09-00794-f001:**
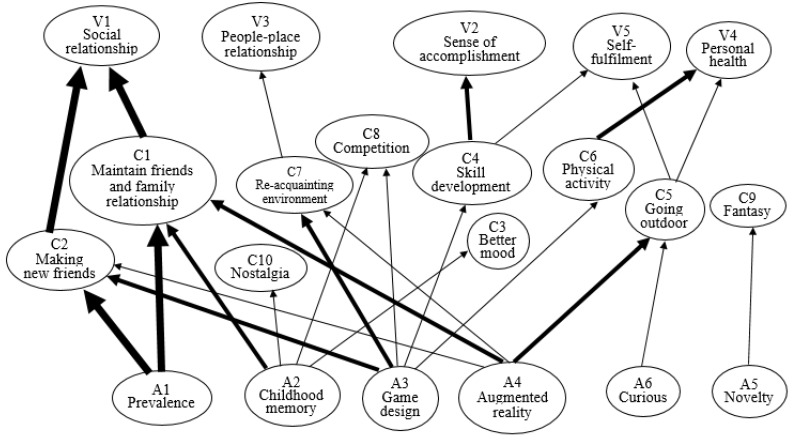
HVM for Pokémon Go experiences.

**Figure 2 healthcare-09-00794-f002:**
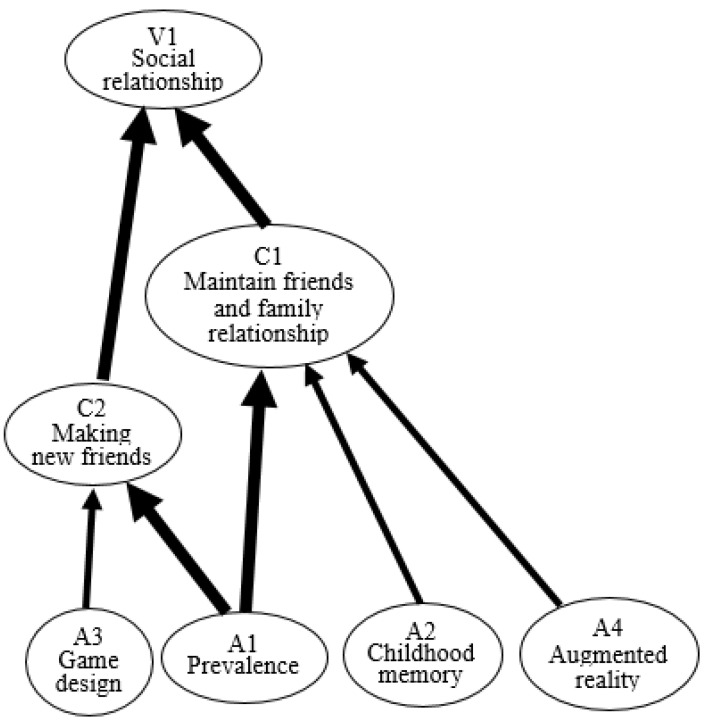
HVM for the social relationship value.

**Table 1 healthcare-09-00794-t001:** Demographics of study participants (N = 34).

Variables	N	Percentage	Variables	N	Percentage
**Gender**			**Occupation**		
Male	21	61.8%	Student	11	32.3%
Female	13	38.2%	Public servant	4	11.8%
**Age**			Laborer	12	35.5%
20–25	11	32.3%	Business owner	4	11.8%
26–30	10	29.4%	Others	3	8.8%
31–35	7	20.6%	**Time usage per day**		
36–40	6	17.7%	Under 60 min	13	38.2%
			61–120 min	6	17.7%
			121–180 min	13	38.2%
			181 min or more	2	5.9%

Note: N = number of participants.

**Table 2 healthcare-09-00794-t002:** Attributes, consequences, and values of Pokémon Go users’ experiences.

Attributes	N	Consequences	N	Values	N
A1 Prevalence	31	C1 Maintaining friends and family relationship	30	V1 Social relationship	26
A2 Childhood memory	28	C2 Making new friends	29	V2 Sense of accomplishment	17
A3 Game design	24	C3 Better mood	17	V3 People–place relationship	16
A4 Augmented reality	19	C4 Skill development	16	V4 Personal health	15
A5 Novelty	15	C5 Going outdoors	16	V5 Self-fulfillment	13
A6 Curiosity	7	C6 Physical activity	15		
		C7 Re-acquainting environment	10		
		C8 Competition	10		
		C9 Fantasy	9		
		C10 Nostalgia	8		

**Table 3 healthcare-09-00794-t003:** The implication matrix for Pokémon Go users’ experiences in this study (N = 34).

Type	C1	C2	C3	C4	C5	C6	C7	C8	C9	C10	V1	V2	V3	V4	V5	Total
A1	30	19	02	01	02	01			01		00; 29	01				57; 29
A2	10	02	05	02		02	01	04		05	00; 03	02; 02			01	34; 05
A3	02	06	02	05	02	05	06	05	01	01		00; 01	00; 03	00; 03		35; 07
A4	08	04		01	07	02	05		01		03	00; 06	00; 02	00; 05	00; 04	31; 17
A5	02	02		02	01		01	01	04						01	14; 00
A6			02	02	03	02	01		01						01; 02	12; 02
C1											31	01		01		33
C2											21	02		01	01	25
C3											02	01		01	02	6
C4											02	09		02	02	15
C5											02		02	03	05	13
C6													02	08	01	11
C7												01	04			5
C8											01	06		01		8
C9												01				1
C10																0
Total	52	33	11	13	15	12	14	10	08	06	62; 32	24; 09	08; 05	17; 08	14; 06	299; 60

Note: 1. The value before semicolon (;) is the frequency of direct connections. 2. The value after semicolon (;) is the frequency of indirect connections.

## Data Availability

Data are not accessible. Accordingly to the informed consent of the study, all participants’ personal information and transcripts have to be confidential. Therefore, data cannot be made publicly available.

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
