# Peer review of "How Do Location-Based Augmented Reality Games Improve Physical and Mental Health? Evaluating the Meanings and Values of Pokémon Go Users’ Experiences through the Means-End Chain Theory"

_healthcare, 2021, doi:10.3390/healthcare9070794_

Round 1

Reviewer 1 Report

This article risks using MEC Theory in a non-traditional setting. It is original for this reason, but it gives the impression that, in some sections, it presents a project and not the results of the research.         For example, in the Introduction (26-60), the literature review (61-181) and in a couple of sections of the Methodology (183-275).

This perception detracts from the quality of the effort made by the authors, so it should be revised, judiciously. Moreover, this has an effect because it forces the question into focus: do augmented reality games improve physical and men's health? Assessing the meanings and values of Pokémon Go users' experiences through media chain theory.

As the result of the survey conducted is that Pokemon players relate the game to health, such observation should be treated correlatively in the title, introduction, methodology, conclusions and bibliography.

1. The title (2-4) is excessively long, perhaps the authors could consider shortening it.  (May be, Assessing the meanings and values of Pokémon Go users' experiences through media chain theory).

2. In item 3.2 (222) on the eligible participants, it would be interesting to know against which options the participants were chosen.

3. In item 3.3. (225) on ethical considerations, it is developed as an administrative technique. It should deal with the ethical dilemmas posed by the games and the values that are presented. 

4. In relation to the bibliography it needs to be updated, out of 52 citations only one is from 2020 (it should be explained in the paper because it has to do with tourism) and only 5 out of 52 are from 2019. They could be useful: Inge Wang (2021) "Systematic literature review on health effects of playing Pokemon Go", and, Mathupayas Thongmak (2020) "Determinants of intention to play Pokemon Go". The rest of the literature, although relevant, gives the impression of being outdated [vr.gr. 2003 (505 and 518), 2005 (522), 2009 (485), 2010 (544), 2011 (514)]. Jonathan Gutman's (1982) classic text "A Means-End Chain Model Based on Consumer Categorization Processes. Journal of Marketing".

Author Response

Reviewer 1

Comments to the Author:

This article risks using MEC Theory in a non-traditional setting. It is original for this reason, but it gives the impression that, in some sections, it presents a project and not the results of the research.  For example, in the Introduction (26-60), the literature review (61-181) and in a couple of sections of the Methodology (183-275).

This perception detracts from the quality of the effort made by the authors, so it should be revised, judiciously. Moreover, this has an effect because it forces the question into focus: do augmented reality games improve physical and men's health? Assessing the meanings and values of Pokémon Go users' experiences through media chain theory.

Response and Changes Made

Thank you for your comments. According to the Healthcare journal’s manuscript preparation in instructions for authors, the manuscript should include Introduction, Materials and Methods, Results, Discussion, Conclusions (optional).

In our opinion, the sections of introduction, literature review, and methodology are necessary for an original paper (Of course, they are not the results). Otherwise, readers cannot understand what the main purpose is in the study (introduction)? What have related study findings found in previous research (literature review)? How to conduct the study to address the purpose of the study.

Furthermore, the means-end chain (MEC) theory is not a risky method. MEC theory is a kind of qualitative method, which has been used to explore individuals’ leisure experiences of attributes, consequences, and values (i.e., Jiang, Scott, & Ding, 2019; Lin, Jeng, & Yeh, 2018). MEC theory can provide insight into why people involve in leisure? What benefits can they be received? and What values they are pursuing? Those are the purposes of the study.

Upon the perception detracts in the paper, please give us some examples, and explain more specifically. Thank you very much!

Lin, C. S., Jeng, M. Y., & Yeh, T. M. (2018). The elderly perceived meanings and values of virtual reality leisure activities: A means-end chain approach. International journal of environmental research and public health, 15(4), 663.

Jiang, S., Scott, N., & Ding, P. (2019). Motivations of experienced leisure travellers: A means-end chain study on the Chinese outbound market. Journal of Vacation Marketing, 25(2), 225-238.

As the result of the survey conducted is that Pokemon players relate the game to health, such observation should be treated correlatively in the title, introduction, methodology, conclusions and bibliography.

  1. The title (2-4) is excessively long, perhaps the authors could consider shortening it.  (May be, Assessing the meanings and values of Pokémon Go users' experiences through media chain theory).

Response and Changes Made

Thank you for your comment. Healthcare journal aims “publishes original theoretical and empirical work in the interdisciplinary area of all aspects of medicine and health care research”. Accordingly, we think that the title of this manuscript should include “How Do Augmented Reality Game Improve Physical and Mental Health”. On the other hand, the issue of improving physical and mental health by AR games is another highlight in the study. According to author’s suggestion, we change the title as “How Do Location-Based Augmented Reality Game Improve Physical and Mental Health? Evaluating the Meanings and Values of Pokémon Go Users’ Experiences through the Means-End Chain Theory”.

  1. In item 3.2 (222) on the eligible participants, it would be interesting to know against which options the participants were chosen.

Response and Changes Made

Thank you for your question. We have stated in the methodology. The study adopted purposive sampling. We have defined the eligible participants, who “have been continuously playing Pokémon Go for over one year, be at least 20 years old (line 223-224).” According to the research topic, exploring the plays’ experiences, the participants must play Pokémon Go continuously. We only recruited participants who are over 20 years old because of ethical considerations. 

  1. In item 3.3. (225) on ethical considerations, it is developed as an administrative technique. It should deal with the ethical dilemmas posed by the games and the values that are presented. 

Response and Changes Made

Thank you for your comment. Ethical considerations are the essential statements in qualitative research. Ethical considerations focus on how to protect participants from potential impacts by involving in the research. For example, confidentiality of personal information. Indeed, it is an administrative technique but essential. We think the ethical dilemmas posed by the games and the values is another study issue, or could you explain more about it? Thank you very much!

  1. In relation to the bibliography it needs to be updated, out of 52 citations only one is from 2020 (it should be explained in the paper because it has to do with tourism) and only 5 out of 52 are from 2019. They could be useful: Inge Wang (2021) "Systematic literature review on health effects of playing Pokemon Go", and, Mathupayas Thongmak (2020) "Determinants of intention to play Pokemon Go". The rest of the literature, although relevant, gives the impression of being outdated [vr.gr. 2003 (505 and 518), 2005 (522), 2009 (485), 2010 (544), 2011 (514)]. Jonathan Gutman's (1982) classic text "A Means-End Chain Model Based on Consumer Categorization Processes. Journal of Marketing".

Response and Changes Made

Thank you for your comment and related reference. We used the No. 49 reference (the 2020 tourism one) to introduce MEC theory and its functions and contributions. Although some of the references are outdated, they are classic and essential to support the study. Thus these references should be kept. We have added the references you provided in the manuscript. Please see the changes below:

Previous studies have examined Pokémon Go users’ motivations and outcomes [6, 8, 31-33]. Several studies have extended existing frameworks to construct a Pokémon Go game-specific motivation scale. For example, Zsila, Orosz, Bőthe, Tóth-Király, Király, Griffiths, and Demetrovics [8] combined qualitative and quantitative methods to extend the Motives for Online Gaming Questionnaire and develop a motivation scale for Pokémon Go. The study identified ten dimensions of motivation – social, escapism, competition, coping, skill development, fantasy, recreation, outdoor activity, nostalgia, and boredom. In addition, several studies have focused on Pokémon Go users’ motivation. Caci, Scrima, Tabacchi, and Cardaci [6] developed their Pokémon Go Motivational Scale through a systematic literature review. The scale consisted of 13 items: curiosity, creativity, different cultures, new people, friends, social experiences, free time, trendiness, escapism, aggression, sexuality, personality, and physical activity. These motivation scales show that Pokémon Go shares some attractive features of traditional online games but is unique in motivating users to get outdoors and engage in physical activity.

Pokémon Go has been proven to bring users mental, social, and physical benefits [6, 31, 34-37]. Tong, Gupta, Lo, Choo, Gromala, and Shaw [31] argued that Pokémon Go could encourage people to go outdoors, engage in positive physical and emotional activities, and explore their surroundings. Many empirical studies have provided evidence that Pokémon Go can effectively promote users’ physical activity and outdoor activity [6, 34, 38]. For example, Nigg, Mateo, and An [34] found that Pokémon Go significantly increased users’ vigorous physical activity by 50 minutes per week and reduced their sedentary activities by 30 minutes per day. However, there is a distinct lack of studies that focus on the specific connections between attributes, consequences, and values in Pokémon Go. As Baranowski [39] proposed that Pokémon Go has potentially powerful ways to increase physical activity. In order to design a more appealing and longer-lasting program, the characteristics and the users’ experiences should be further understood.  By exploring these relationships, we can not only clarify why people play Pokémon Go, what benefits they receive from playing, and what players value about these games – we can also deepen our understanding of Pokémon Go apply this understanding to unlock the educational and therapeutic potential of Pokémon Go.

  1. Thongmak, M., Determinants of intention to play Pokémon Go. Heliyon 2020, 6 (12), e03895.

35. Wang, A. I., Systematic literature review on health effects of playing Pokémon Go. Entertain. Comput 2021, 38, 100411.

Reviewer 2 Report

Dear editor and authors, thank you very much for the opportunity to review this manuscript.

This paper presents qualitative data that describes how Pokémon Go can be considered an active leisure.

The paper is original, well written, and the design is appropriate as well as the results are clearly described. However, despite these strengths, I have some theoretical concerns/questions that I tried to summarize in the following comments for authors, and that limit the contribution of the study to the field. I hope the authors will find these comments helpful. Consequently, in my opinion, the manuscript should be considered for publication in the journal after review these issues in the discussion section.

Comments for authors:

1 – I do not understand Table 1. Results here are not clear.

2 – Limitations of the study should be expanded in an independent section. I recommend a deep lecture of the following papers:

  • Gabbiadini, A., Sagioglou, C., & Greitemeyer, T. (2018). Does Pokémon Go lead to a more physically active life style?. Computers in Human Behavior, 84, 258-263.

  • Kim, Y., Bhattacharya, A., Kientz, J. A., & Lee, J. H. (2020, April). " It Should Be a Game for Fun, Not Exercise": Tensions in Designing Health-Related Features for Pokémon GO. In Proceedings of the 2020 CHI Conference on Human Factors in Computing Systems (pp. 1-13).

My main concern is the assumption that playing Pokemon Go improves health. This is no right. Technology can’t never replace physical exercise or sport benefits. Maybe can be used as a leisure activity but we need to control carefully other potential hazards as gaming obsession, social anxiety and not confound connection of in-real-life experiences with the game vs. different individual contexts. In certain ages this could be really important (children and teenagers).

3 – Introduce in the conclusions section some of these ideas.

4 – Could you cite any article to justify the sample size? You use a qualitative element of research to set the parameters for a further, positivist quantification. This means that you apply a positivist approach to qualitative research and, under this approach, a criticism of sample size because of smallness may well be justified. This is because the qualitative sample size has to be representative of the population under consideration as a breadth of inquiry is anticipated.

               4.1. What about the difference in sample between male (62%) and female (38%).

The paper is well presented and I recommend it for publication after the previous suggestions. Thank you again to the editor and authors for the possibility to review this manuscript.

Author Response

Reviewer 2

Dear editor and authors, thank you very much for the opportunity to review this manuscript.

This paper presents qualitative data that describes how Pokémon Go can be considered an active leisure.

The paper is original, well written, and the design is appropriate as well as the results are clearly described. However, despite these strengths, I have some theoretical concerns/questions that I tried to summarize in the following comments for authors, and that limit the contribution of the study to the field. I hope the authors will find these comments helpful. Consequently, in my opinion, the manuscript should be considered for publication in the journal after review these issues in the discussion section.

Response and Changes Made

Thank you very much for your kind words and efforts. Your comments and suggestions are helpful for improving our manuscript! Please have a look at our responses and the change made.

Comments for authors:

1 – I do not understand Table 1. Results here are not clear.

Response and Changes Made

Thank you for pointing out the defects! We added the number of participants (N) in Table 1, presented the demographic variable with bold font, and added the bottom line to divide variables. Please see the changes below:

Table 1. Demographics of study participants (N=34).

Variables

N

Percentage

Variables

N

Percentage

Gender

Occupation

  Male

21

61.8%

  Student

11

32.3%

  Female

13

38.2%

  Public servant

4

11.8%

Age

  Labourer

12

35.5%

  20-25

11

32.3%

  Business owner

4

11.8%

  26-30

10

29.4%

  Others

3

  8.8%

  31-35

7

20.6%

Time usage per day

  36-40

6

17.7%

  Under 60 minutes

13

38.2%

  61-120 minutes

6

17.7%

  121-180 minutes

13

38.2%

  181 minutes or more

2

  5.9%

Note: N=Number of participant

2 – Limitations of the study should be expanded in an independent section. I recommend a deep lecture of the following papers:

Gabbiadini, A., Sagioglou, C., & Greitemeyer, T. (2018). Does Pokémon Go lead to a more physically active life style?. Computers in Human Behavior, 84, 258-263.

Kim, Y., Bhattacharya, A., Kientz, J. A., & Lee, J. H. (2020, April). " It Should Be a Game for Fun, Not Exercise": Tensions in Designing Health-Related Features for Pokémon GO. In Proceedings of the 2020 CHI Conference on Human Factors in Computing Systems (pp. 1-13).

My main concern is the assumption that playing Pokemon Go improves health. This is no right. Technology can’t never replace physical exercise or sport benefits. Maybe can be used as a leisure activity but we need to control carefully other potential hazards as gaming obsession, social anxiety and not confound connection of in-real-life experiences with the game vs. different individual contexts. In certain ages this could be really important (children and teenagers).

Response and Changes Made

Thank you for your suggestions and papers! We have expanded the limitations of the study in an independent section. Please see the changes below:

  1. Limitations of the study

This study examined Pokémon Go users’ experiences and attempted to identify significant value chains among these experiences. It has a few limitations. First, this study is limited to Taiwanese users. Although other, similar studies have been conducted in different contexts, the limitations of this study mean that it is difficult to generalise its results. Future studies should collect data from many more countries, including Western countries. Second, this study adopted the soft laddering method to construct HVM to model Pokémon Go users’ experiences. However, there is still a lack of quantitative data verifying the HVM’s structure. Therefore, the hard laddering method could be useful in future empirical studies.

Upon your main concern, I agreed that “technology can’t never replace physical exercise or sport benefits”. However, we think technology can “assist” people in increasing physical exercise and indirectly receiving sport benefits. Of course, controlling potential hazards from technology/games is very important. Gabbiadini’s (2018) study indicated a positively causal relationship between Pokémon Go related physical activity and overall physically active behavior, despite the researchers arguing that the apps mere adoption does not reliably change people's behavior in general. Indeed, Pokémon Go may not improve people’s physically active behavior in general, but it can force people, or those who may have a sedentary lifestyle and interpersonal constraints, to start their physical activity and social interaction. We also believe technology/games to children’s and teenagers’ physical, mental, and social health is essential. We think this is an excellent topic in future studies.

3 – Introduce in the conclusions section some of these ideas.

 Response and Changes Made

Thank you for your comments! We added some arguments in the conclusions section. Please see the changes below:

In short, Pokémon Go brings various benefits to its users. Although, as mentioned, the literature has indicated that traditional online games can improve users’ mental health, Pokémon Go’s game design and incorporation of location-based AR technology mean that it provides more physical and social benefits to users than its traditional online counterparts. The location-based AR technology and game design of Pokémon Go have been shown to avoid the usual lack of physical activities and sedentary lifestyle associated with online games, as well as to increase outdoor physical activity [6, 8, 34-35]. Although Pokémon Go cannot replace physical exercise in general and receive sport benefits directly, it assists people in starting their physical activity and social interaction. Accordingly, this study proposes that Pokémon Go is not treated as pure entertainment, but as a potential tool for promoting public health, well-being, and outdoor exploration [54-55]. In short, it can be considered as an instrument of public well-being promotion [54].

4 – Could you cite any article to justify the sample size? You use a qualitative element of research to set the parameters for a further, positivist quantification. This means that you apply a positivist approach to qualitative research and, under this approach, a criticism of sample size because of smallness may well be justified. This is because the qualitative sample size has to be representative of the population under consideration as a breadth of inquiry is anticipated.

Response and Changes Made

Thank you for your comment. We added the statement and reference to support sample size decision. Please see the changes below:

In order to enhance the validity of data in this study, the authors employed purposive and snowball sampling to select 34 eligible study participants. Eligible participants have been continuously playing Pokémon Go for over one year, be at least 20 years old, and voluntarily submit themselves to the 30–40 minute interviews described above. According to the above principles, eligible participant recruitment was divided into two stages. In the first stage, 10 and 13 eligible participants were recruited from National Don Hwa University and Qixingtan Beach, respectively – two popular spots for Pokémon Go activity in Hualien County, Taiwan. In the second stage, the remaining 11 eligible participants were recruited from the initial participants’ acquaintances. Interviews were conducted face-to-face or online depending on the interviewees’ preferences. Sample size of the study was decided by data saturation. The data collection was stopped at number 34 participants because no additional data could be established to develop new properties of categories[51].

4.1. What about the difference in sample between male (62%) and female (38%).

 Response and Changes Made

Thank you for your question. Did you mean the different findings between the male and female? If yes, unfortunately, it is out of scope in the study. But it is worthy to explore in future studies. 

Reviewer 3 Report

This is an informative and interesting study.

There are some minor grammatical errors that should be cleared up, but these do not detract from the quality and importance of the research at all. 

Author Response

Reviewer 3

This is an informative and interesting study.

There are some minor grammatical errors that should be cleared up, but these do not detract from the quality and importance of the research at all. 

 Response and Changes Made

Thank you for your kind words and encouragement. We have revised the grammatical errors carefully. For example, 

Numerous studies have verified that online games can negatively influence peoples’ well-being [13-14].

However, location-based augmented reality (AR) technology has brought gamers a whole new interface, playing games, and a set of benefits.

Reviewer 4 Report

This paper analyses the motivation of a group of Taiwanese users of Pokémon Go to play this AR game through the means-end chain theory.

The main innovation of the paper is the use of the means-end chain theory to evaluate the motivations to play this AR game.

However, this study involves a small number of participants (only 34) when compared to previous ones, which means that it cannot be considered representative of the Taiwanese Pokémon Go users.

In section 2.2, the authors say that:

“For this reason, the main differentiation between traditional online games and AR games is the fact that the latter allow players to participate in gaming experiences outdoors.”

However, not all AR games allow players to participate in gaming experiences outdoors. Only the AR games that integrate users’ locations via Global Positioning System (GPS) technology are designed for gaming experiences outdoors.

So, it is important to clarify that exist several types of AR games.

The results presented are in line with previous studies, it found that users value the social relationships that Pokémon Go facilitates between friends, family, strangers, and neighbours, and extends online social interactions, enriching both online and offline relationships. 

So, it provides more physical and social benefits to users than its traditional online counterparts.

The AR technology and game design of Pokémon Go have been shown to avoid the usual lack of physical activities and sedentary lifestyle associated with online games, as well as to increase outdoor physical activity and social interactions.

In short, despite the small number of participants in the study, it uses a new methodology to analyse the motivations of Pokemon Go players, and given the pandemic situation, I think that this paper can be accepted after minor changes.

Some mistakes in text:

- Some references in the text appear with three or four authors, while others appear with the first author et al. (for example, lines 73 and 82 vs 140 and 175)

Please be consistent in the references.

- In line 316 appears " In the matrix, ..." but it must be "In Figure 1, ..."

- Correct the name of the first author in reference 47.

Some references that are missing:

- Juho Hamari, Aqdas Malik, Johannes Koski & Aditya Johri (2019) “Uses and Gratifications of Pokémon Go: Why do People Play Mobile Location-Based Augmented Reality Games?”, International Journal of Human–Computer Interaction, 35:9, 804-819, DOI: 10.1080/10447318.2018.1497115

- Janne Paavilainen, Hannu Korhonen, Kati Alha, Jaakko Stenros, Elina Koskinen, Frans Mayra (2017) “The Pokémon GO Experience: A Location-Based Augmented Reality Mobile Game Goes Mainstream”, Proceedings of the 2017 CHI Conference on Human Factors in Computing Systems, May 2017, Pages 2493–2498, https://doi.org/10.1145/3025453.3025871 

Author Response

Reviewer 4

This paper analyses the motivation of a group of Taiwanese users of Pokémon Go to play this AR game through the means-end chain theory.

The main innovation of the paper is the use of the means-end chain theory to evaluate the motivations to play this AR game.

However, this study involves a small number of participants (only 34) when compared to previous ones, which means that it cannot be considered representative of the Taiwanese Pokémon Go users.

Response and Changes Made

Thank you for your comment. We added the statement and reference to support sample size decision. Please see the changes below:

In order to enhance the validity of data in this study, the authors employed purposive and snowball sampling to select 34 eligible study participants. Eligible participants have been continuously playing Pokémon Go for over one year, be at least 20 years old, and voluntarily submit themselves to the 30–40 minute interviews described above. According to the above principles, eligible participant recruitment was divided into two stages. In the first stage, 10 and 13 eligible participants were recruited from National Don Hwa University and Qixingtan Beach, respectively – two popular spots for Pokémon Go activity in Hualien County, Taiwan. In the second stage, the remaining 11 eligible participants were recruited from the initial participants’ acquaintances. Interviews were conducted face-to-face or online depending on the interviewees’ preferences. Sample size of the study was decided by data saturation. The data collection was stopped at number 34 participants because no additional data could be established to develop new properties of categories[51].

In section 2.2, the authors say that:

“For this reason, the main differentiation between traditional online games and AR games is the fact that the latter allow players to participate in gaming experiences outdoors.”

However, not all AR games allow players to participate in gaming experiences outdoors. Only the AR games that integrate users’ locations via Global Positioning System (GPS) technology are designed for gaming experiences outdoors.

So, it is important to clarify that exist several types of AR games.

Response and Changes Made

Thank you for your comment. We have emphasized the Pokémon Go is a kind of “location-based AR game” in the manuscript. For example, “Pokémon Go is one of the most popular location-based AR games in the world. As part of the hugely popular Pokémon series of games, it integrates users’ locations via Global Positioning System (GPS) technology, view (via their camera), and virtual information – the various wandering creatures of the Pokémon universe – with users’ real, immediate environment”.

We also made the same change in other sections. Please see the highlight parts with yellow in the manuscript. 

The results presented are in line with previous studies, it found that users value the social relationships that Pokémon Go facilitates between friends, family, strangers, and neighbours, and extends online social interactions, enriching both online and offline relationships. 

So, it provides more physical and social benefits to users than its traditional online counterparts.

The AR technology and game design of Pokémon Go have been shown to avoid the usual lack of physical activities and sedentary lifestyle associated with online games, as well as to increase outdoor physical activity and social interactions.

In short, despite the small number of participants in the study, it uses a new methodology to analyse the motivations of Pokemon Go players, and given the pandemic situation, I think that this paper can be accepted after minor changes.

Response and Changes Made

Thank you very much for your comments and kind words.

Some mistakes in text:

- Some references in the text appear with three or four authors, while others appear with the first author et al. (for example, lines 73 and 82 vs 140 and 175)

Please be consistent in the references.

- In line 316 appears " In the matrix, ..." but it must be "In Figure 1, ..."

- Correct the name of the first author in reference 47.

Some references that are missing:

- Juho Hamari, Aqdas Malik, Johannes Koski & Aditya Johri (2019) “Uses and Gratifications of Pokémon Go: Why do People Play Mobile Location-Based Augmented Reality Games?”, International Journal of Human–Computer Interaction, 35:9, 804-819, DOI: 10.1080/10447318.2018.1497115

- Janne Paavilainen, Hannu Korhonen, Kati Alha, Jaakko Stenros, Elina Koskinen, Frans Mayra (2017) “The Pokémon GO Experience: A Location-Based Augmented Reality Mobile Game Goes Mainstream”, Proceedings of the 2017 CHI Conference on Human Factors in Computing Systems, May 2017, Pages 2493–2498, https://doi.org/10.1145/3025453.3025871 

Response and Changes Made

Thank you for pointing out these defects. We have corrected the defects following your comments. For example,

Tong, Gupta, Lo, Choo, Gromala, and Shaw [2] argued that Pokémon Go could encourage people to go outdoors, engage in positive physical and emotional activities, and explore their surroundings.

Nigg, Mateo, and An [5] found that Pokémon Go significantly increased users’ vigorous physical activity by 50 minutes per week and reduced their sedentary activities by 30 minutes per day.

The same defects have been corrected. Please see the highlight parts with yellow in the manuscript.

The words “In the matrix” have been change into “In Figure 1”.

The matrix displays the frequency of direct and indirect linkages between attributes, consequences, and values. This study recorded a total of 299 direct linkages and 60 indirect linkages. In Figure 1, a line’s thickness represents the number of linkages, which shows the different levels of significant connections. For example, the thickest lines represent more than ten linkages, the next-thickest line between six and nine linkages, and the thinnest line less than five linkages. Using this matrix, the authors then constructed the HVM for Pokémon Go users’ experiences, displaying 25 total linkages – four powerful connections, seven strong-to-average connections, and 14 fewer strong connections. The overall HVM is separated into three sub-themes to illustrate Pokémon Go users’ experiences.

Reference 52 of the author’s name, tom Dieck, is correct.

Two referenecs have added in the manuscript.

  1. Hamari, J.; Malik, A.; Koski, J.; Johri, A., Uses and gratifications of pokémon go: Why do people play mobile location-based augmented reality games? International Journal of Human–Computer Interaction 2019, 35 (9), 804-819.

  1. Paavilainen, J.; Korhonen, H.; Alha, K.; Stenros, J.; Koskinen, E.; Mayra, F. In The Pokémon GO experience: A location-based augmented reality mobile game goes mainstream, Proceedings of the 2017 CHI conference on human factors in computing systems, 2017; pp 2493-2498.

Round 2

Reviewer 1 Report

Dear Authors: I appreciate the response to my comments, which only tried to contribute to improve the perception of your work, which has my recognition. I have taken into account your explanatory arguments, the defense of some formal issues and the improvements you have made to the text; they are quite satisfactory and the positive evaluation of your work can be understood as my agreement with your explanations, defenses and improvements. 

I thank all of you for asking me some questions, in particular the question about ethical considerations, because indeed, as you point out, addressing them would give for another paper. However, I believe that in our research we could understand ethics not only as a procedure but also as a behavior.
Best regards.